# Development and Validation of a Short Sport Nutrition Knowledge Questionnaire for Athletes

**DOI:** 10.3390/nu12113561

**Published:** 2020-11-20

**Authors:** Karla Vázquez-Espino, Carles Fernández-Tena, Maria Antonia Lizarraga-Dallo, Andreu Farran-Codina

**Affiliations:** 1Department of Nutrition, Food Science and Gastronomy, XaRTA–INSA, Faculty of Pharmacy, University of Barcelona, Av. Prat de la Riba Campus de l’Alimentació de Torribera, 171, Santa Coloma de Gramenet, E-08921 Barcelona, Spain; karlavespino@gmail.com (K.V.-E.); mlizarraga@ub.edu (M.A.L.-D.); 2Independent Researcher, Pau Claris 165, 3-B, E-08037 Barcelona, Spain; carlos.fernandez.tena@gmail.com; 3FC Barcelona Medical Services, Avda. Onze de Setembre, s/n, Sant Joan Despí, E-08970 Barcelona, Spain

**Keywords:** sports nutrition, nutrition knowledge, validity, reliability, classical test theory, item response theory, Rasch model, athletes

## Abstract

Weak evidence exists on the relationship between nutritional knowledge and diet quality. Many researchers claim that this could be in part because of inadequate validation of the questionnaires used. The aim of this study was to develop a compact reliable questionnaire on nutrition knowledge for young and adult athletes (NUKYA). Researchers and the sport clubs medical staff developed the questionnaire by taking into consideration the latest athlete dietary guidelines. The questionnaire content was validated by a panel of 12 nutrition experts, and finally tested by 445 participants including athletes (*n* = 264), nutrition students (*n* = 49) and non-athletes with no formal nutrition knowledge (*n* = 132). After consulting the experts, 59 of the 64 initial items remained in the questionnaire. To collect the evaluation of experts, we used the content validity index, obtaining high indices for relevance and ambiguity (0.99) as well as for clarity and simplicity (0.98). The final questionnaire included 24 questions with 59 items. We ensured construct validity and reliability through psychometric validation based on the Classical Test Theory and the Item–Response Theory (Rasch model). We found significant statistical differences comparing the groups of nutrition knowledgeable participants with the rest of the groups (ANOVA *p* < 0.001). We verified the questionnaire for test–retest reliability (*R* = 0.895, *p* < 0.001) and internal consistency (Cronbach’s α=0.849). We successfully fit the questionnaire data to a rating scale model (global separation reliability of 0.861) and examined discrimination and difficulty indices for items. Finally, we validated the NUKYA questionnaire as an effective tool to appraise nutrition knowledge in athletes. This questionnaire can be used for guiding in educational interventions, studying the influence of nutrition knowledge on nutrient intake and assessing/monitoring sport nutritional knowledge in large groups.

## 1. Introduction

The relationships among knowledge, attitudes and behaviors regarding nutrition are not fully understood. However, nutrition knowledge is considered necessary to incorporate healthier food habits, and recent work indicates that it may play a small but pivotal role in this process [1]. There is no evidence sustaining strong associations among the above-mentioned factors. Many experts claim that this is because of the systematic inadequacy of the validation methods in the field [2,3,4,5,6]. Therefore, it is necessary to develop validated nutritional questionnaires to correctly assess the possible connection between behavior and nutrition knowledge.

Among the general population, attitude and education are known important factors for sensible nutritional choices [7,8]. Professional athletes who already recognize that such choices can strongly affect their performance. For them, the key factors are nutrition knowledge, followed by attitude and sources of nutritional information. Moreover, nowadays, athletes reach the professional level at a younger age than before, and this makes them more vulnerable to poor diet choices [2]. In the context of elite sports, sport-specific questionnaires are the preferred tool to assess nutrition knowledge, since the questions are pertinent to the demands of the type of sport. Additionally, in this context, the correct assessment of nutrition knowledge is particularly important since this is a key modifiable variable that can affect athlete performance [9].

The latest systematic review [4] showed that most questionnaires on nutrition knowledge for athletes and coaches tested outdated recommendations, missed comprehensiveness, failed to cultural suitability and needed an adequate validation. Several recent works improved the quality of methodologies, although they were restricted to specific sports and had a very limited sample size [10,11,12,13]. Furthermore, most of these studies superficially rely on the subjectivity of experts’ panels to assess the validity of the content [10,14], and do not cover multilingual scenarios. Some of them compromised the quality of responses by distributing the questionnaires online to athletes, without any supervision [14,15,16]. This might have allowed for possible sharing and discussion of the responses among participants. Finally, many validated questionnaires include lots of items and require time to be completed. Thus, there is still work to be done for improving assessment tests on nutrition knowledge.

The aims of this study were: (i) to develop a compact reliable knowledge questionnaire on nutrition knowledge for young and adult athletes (NUKYA) for assessing the main bases of nutrition knowledge specifically in sports team; and (ii) to psychometrically validate the questionnaire.

## 2. Materials and Methods

### 2.1. Study Design

This instrument validation study was conducted in sport clubs and main entities involved. Recruitment of participants and data collection occurred between January 2015 and July 2017. This research was approved by the ethics committee for Clinical Research of the Catalan Sports Council 18/2019/CEICEGC.

### 2.2. Study Development

We intended to measure the nutrition knowledge of young and adult athletes members of elite sport teams. We aimed to develop a compact questionnaire for these athletes that generally follow varying diets throughout a regular season, depending on their sport and individual needs. We started by defining the areas of knowledge to evaluate in the test [17]. According to the educational objectives of athletes who were part of sports clubs, we established the following sections: *macronutrients*, *micronutrients*, *hydration* and *food intake periodicity*. We discarded other commonly analyzed topics such as *supplements*, *weight management* and *alcohol* to prioritize a compact version. Furthermore, some of the discarded topics should have been subjected to advice by a nutritionist or a medical doctor, and others are discouraged or forbidden for athletes (e.g., alcohol).

#### 2.2.1. Generation of Items

Defining the precise pool of questions is fundamental to cover the selected topics and make relevant measurements [18,19]. Initially, redundancy helped ensure the coverage; subsequently, we discarded out-of-scope questions to grant relevance. More than 80 items were conceptualized. Afterwards, researchers and experts from the medical team of the sports club thoroughly performed a screening process. They selected 64 multiple choice questions and distributed them among areas of interest, to give a synthesized appearance. We included the answer *“not known/not sure”* to avoid the participants from guessing and distinguish between possibly forgotten or totally unknown items. A negative scoring of $\frac{1}{n-1}$ was applied to wrong answers, where *n* is the number of options included in the test item. A positive score of 1 was given to correct answers [20]. This served to mitigate the effect of lucky correct answers. To achieve a multilingual version, we originally wrote the questions in Spanish and later translated them into English, under rigorous review by multiple native speakers.

#### 2.2.2. Content Validity

The content of the questionnaire was validated after being reviewed by a panel of experts who assessed its accuracy by using a qualitative approach. The panel was formed by 12 international experts (5 English native speakers and 7 Spanish native speakers) with multiple career backgrounds. Each of them had more than a decade of experience in sports. To assess the questionnaire, panelists received information on general and specific topics via e-mail. In addition, they were asked to complete a rating response form. All the questionnaire items were evaluated by each member of the panel through a 4-point Likert scale under the following criteria: relevance, *1 = ‘not relevant’ to 4 = ‘truly relevant’*; clarity, *1 = ‘not clear’ to 4 = “very clear’*; simplicity, *1 = ‘not simple’ to 4 = ‘easy to understand’*; and ambiguity, *1 = ‘doubtful’* to 4 = ‘clear’. For each item, we calculated the content validity index (I-CVI) by dividing the number of experts who gave a rating of either *3* or *4* by the total number of experts. We finally computed the content validity score of the whole scale (S-CVI) by adding all the items rated *3* and *4* by all the experts. In addition, the experts had the opportunity to share their comments on each item. This contributed to assessing the *face-validity* of the questionnaire, answering the question whether individuals (experts or respondents) believe that the questionnaire measures what it is intended to measure [21]. After assessing all these results (I-CVI, S-CVI and face-validity), we reformulated certain questions following the comments and face-validity insights from the panel. We finally removed the questions that did not have a I-CVI ≥ 0.78, as commonly defined for adequate content validity [22].

#### 2.2.3. Pilot Study and Item Wording Improvement

To evaluate the practical aspects of the questionnaire and assure its feasibility (vocabulary, phrasing, clarity of the instructions, wording mistakes and resolution time) and face-validity, we conducted a pilot test. Vocabulary and phrasing were especially important because nutritional knowledge of athletes is associated with their age, and respondents must understand all the concepts in the questionnaire. The pilot group came from the minor divisions of a professional football club. It included athletes whose range of age (12–15 years) was similar to the minimum age of the population for which the questionnaire was intended. The pilot test was delivered in a written form, encouraging participants to freely address any doubt or question orally or in writing. We found that participants at the age of 12 lacked the appropriate vocabulary to understand some items. Hence, some items were adapted for an easier understanding and the minimum age for the test was set to 13 years.

#### 2.2.4. Construct Validity

To assess discriminant validity, the questionnaire was administered to several groups with different nutrition training, age and formal education. Discriminant validity is a psychometric characteristic related to construct validity, and we expected its score to be different between the groups. Four groups were formed according to their predictable nutrition knowledge: (i) elite athletes with non-formal education (FCB, players from sport clubs); (ii) non-athletes with formal education in nutrition (UB-NHD, students enrolled in the last year of nutrition degree); (iii) non-athletes university students non-formally educated in nutrition (UB-FIL, students in philosophy at the UB); and (iv) non-athletes high school students non-formally educated in nutrition (CSK, high school students at a private school). The differential factor to create Groups (iii) and (iv) was to accommodate a similar age spectrum from Groups (i) and (ii), respectively. The groups were recruited following the best practices from well-known entities, using purposive sampling agreements.

#### 2.2.5. Reliability

Test–retest reliability was measured by providing the same questionnaire to all the groups within a time-lapse of more than two weeks and less than four. To study the regularity of the questionnaire stability between groups, we calculated the correlation between test and retest data for each subject. We also assessed the data for internal consistency by using the test data set. Finally, to identify meaningful patterns, we analyzed the questionnaire items grouping them by sections.

#### 2.2.6. Sample Size Calculation

To determine the adequate sample size for mean comparison between groups, we conducted a statistical power analysis. We used the G-Power software [23], considering a two-tail independent sample *t*-test, using estimations of means and standard deviations from previous pilot data. We finally adopted common values for the significance level (α=0.05) and a high statistical power (1 − β=0.95). All the subsequent tests had samples sizes based on the conducted power analysis that yielded to a minimum of 33 participants per group. This number is mostly covered by the sample sets that were used throughout this work, with an estimated effect size of d=0.90.

#### 2.2.7. Statistical Analysis

We assessed normality of the data through the Shapiro–Wilk test and checked homoscedasticity through the Levene’s test. We used parametric tests (independent or dependent samples t-test, ANOVA and Pearson correlation) for normal data. When assumption of normality was not possible, we used non-parametric equivalents of the above-mentioned tests (e.g., Mann–Whitney U test, Kruskal–Wallis and Spearman’s rank correlation coefficient). We evaluated internal consistency using Cronbach’s α and split-half estimate, followed by Spearman–Brown’s ρ. Finally, we calculated item difficulty and discrimination indices according to Muñiz [20].

#### 2.2.8. Rasch Analysis

We used Rasch models to measure questionnaire responses as a trade-off between the difficulty of the items and the abilities and attitudes of respondents [24]. In our case, we chose to work with a Rating Scale Model [25]. This is an early polytomous extension of the original Rasch model and one of the most widely employed models in item response theory. Rating scale models required categorical data as input. Therefore, we converted the continuous score scale originally used in the study into three possible discrete categories: wrong answers, *“not known/not sure”* answers and correct answers. The objective of our Rasch analysis was to evaluate the quality of the measurement, through a series of crucial assumptions that were empirically testable (unidimensionality, conditional independence and monotonicity). Finally, we wanted to identify questionnaire items that did or did not conform to the model. The results of such analysis allowed us to determine which questionnaire items could be irrelevant or out of scope for the targeted construct.

## 3. Results

### 3.1. Subjects

The characteristics of subjects that participated in the reliability and validation study of this questionnaire are described in Table 1. The four groups considered were elite athletes (FCB), final year students of nutrition (UB-NHD), high-school students (CSK) and second year students of philosophy (UB-FIL). Most of the participants were Spanish (92%) and the rest were from 22 different countries in Europe, South America and Africa. Ninety percent of foreign participants were athletes from sports club. The group of athletes (FCB) came from the soccer, basketball, futsal and roller hockey teams.

### 3.2. Content Validity, Feasibility and Face-Validity

For both versions of the questionnaire in Spanish and English, I-CVI and S-CVI indices were computed from the ratings assigned to each item in the questionnaire by seven Spanish-speaking and five English-speaking experts, respectively. Table 2 shows the resulting I-CVI and S-CVI scores for the initial and final set. Two out of the 26 initial questions were removed, as they were, respectively, rated 0.71 for I-CVI on clarity and simplicity. Five items were reformulated based on the expert feedback. The final questionnaire consists of 24 questions with 59 items and was completed by respondents in 12 min on average. S-CVI scores in the reviewed item set were above 0.90 for relevance and ambiguity and almost 0.90 for clarity and simplicity [26]. The questionnaire was considered to adequately assess the nutritional knowledge in the sections included (macronutrients, micronutrients, hydration and food intake periodicity) by respondents of the pilot test and by the experts of the panel. Appendix A shows information about the content and question response categories.

### 3.3. Construct Validity

#### 3.3.1. Discriminant Validity

Since the questionnaire was administered to several groups with different nutrition training, we expected significantly different scores between some of them. To test if these differences were coherent and statistically significant, the scores from the test set (Table 3) were compared between the four groups. Previously, we checked data normality in each group and homoscedasticity using Shapiro–Wilk and Levene tests. Shapiro–Wilk indicated normality in the four groups (*p* > 0.05), but variances were not homogeneous. This was basically due to the low variance detected in the UB-NHD group: when the Levene test (using the STATA robvar procedure) was performed on the three remaining groups, the null hypothesis (presence of variance homogeneity) could not be refused. Additionally, homoscedasticity is an important condition only when sample size in each group is *n* ≤ 30 [27]. Thus, we ran one-way ANOVA to detect the presence of differences between groups in the test score, with Bonferroni *post-hoc* test for multiple pairwise comparisons.

The one-way ANOVA test showed statistically significant differences between the four groups (F=178.56,p<0.00001). The Bonferroni test detected statistically significant differences between UB-NHD group, in which the best test performance was expected, and the rest of the groups. In addition, the UB-FIL group was significantly different from both the CSK and FCB groups. The *p*-value was lower than 0.0001 in all cases, except between UB-FIL and CSK where *p* = 0.035. These results are coherent with what expected according to the education and the age of participants.

#### 3.3.2. Item Difficulty and Discrimination

We calculated the difficulty index $d_{i}\in \left[0,1\right]$ for each item *i* in the questionnaire, by dividing the number of participants that gave the correct answer by the total number of respondents, i.e., $d_{i}=N_{correct}/N_{total}$. Items with an associated index below 20% or over 80% were considered for removal, as they could be excessively difficult and excessively easy, respectively. Only six items fell in these categories. Additionally, we computed the discrimination index $D_{i}\in \left[-1,1\right]$ as $D_{i}=\left(N_{>median}^{correct}-N_{<median}^{correct}\right)/\left(N_{total}/2\right)$ to assess the degree that each question discriminates high-performance respondents from low-performance ones. We confirmed that none of the questions led to an erroneous discrimination, as all indices were positive. Figure 1 shows bar plots of the item difficulty indices and the item discrimination indices.

#### 3.3.3. Sectional Analysis

We divided the scores from the original test set into sections for a more detailed analysis. The 59 questionnaire items were distributed per topics into *macronutrients* ($n_{S}=30$), *micronutrients* ($n_{S}=19$), *hydration* ($n_{S}=7$) and *food intake periodicity* ($n_{S}=3$). Each score *C* was normalized into $C^{\prime }$ based on its maximum score, and negative values were clamped to zero, i.e., $C^{\prime }=\max\left(0,100\times C/n_{C}\right)$. Table 4 contains the measures of center and spread for each normalized section, computed for each subject group and globally. As expected, the UB-NHD group obtained the bests scores in all four sections. The FCB group obtained better results than CSK and UB-FIL groups in micronutrients, hydration and food intake periodicity sections. The CSK group was the one with the lowest scores. It should be noted that the FCB group, which has a similar age profile as the CSK group (Table 4), regularly received information about hydration and food intake periodicity, as reflected by the data.

#### 3.3.4. Rasch Analysis

We used the R package eRm (extended Rasch modeling) for fitting of data into the rating scale model and analyze them. We categorized the response of each participant (*n* = 445, see Table 1) into *wrong*/*do not know*/*correct* (0/1/2, respectively), forming an input data matrix of 59 × 414. The model was fitted to the data with respect to three main dimensions: knowledge on macronutrients, knowledge on micronutrients and knowledge on food intake periodization. We considered periodization for both liquids and solids, therefore including hydration items.

We fitted the model to the data on the knowledge for each of the three above mentioned dimensions. To assess goodness-of-fit of the model, we plotted person–item maps, conducted Wald tests on item level and obtained reliability scores for each dimension. Table 5 shows the reliability measures obtained per dimension and Figure 2 shows a spatial representation of the questionnaire items on the latent dimension. Separation reliability approximately measures the ratio between true variance and observed variance, i.e., the Rasch depicts the Cronbach’s alpha. The three constructs yielded reliabilities over 60%, and the highest value was obtained for macronutrients. We observed the largest mean square measurement error for food intake periodization.

Regarding Wald tests, we computed test statistics and *p*-values for each element in the questionnaire. We made calculations for Categories 1 (*“not known/not sure”*) and 2 (correct) with respect to 0 (incorrect). Within the second category, the majority of items (54 out of 59) presented *p*-values much below the significance level of 0.05, with few exceptions: (i) Item 5.6 (proteins in dry fruit); (ii) Items 10.b, 10.c and 10.d (hydration); and (iii) Item 13 (isotonic drinks). Thus, these items could be removed without affecting the model; however, they were retained because they provided critical information about athletes’ knowledge on how to identify foods containing proteins or on their hydration status. As for Category 1, a higher percentage of items presented *p*-values below 0.05.

### 3.4. Reliability

#### 3.4.1. Internal Consistency

To evaluate the internal consistency of the questionnaire, we calculated Cronbach’s alpha. We chose the unstandardized variant, as all items share an equal score range. The input matrix of 59 test item scores by 442 respondent participants (the above-mentioned groups and an additional group of 268 athletes) produced an alpha of 0.849. This indicates a good item consistency for our unidimensional latent construct (i.e., nutritional knowledge). We repeated the same computation over the retest scores, where data from 173 respondents were available, resulting in alpha = 0.887. We also evaluated the effect of removing a single item from the test. In this case, the modified alphas ranged within $\left[0.840,0.856\right]$, barely 1% lower than the original value. Therefore, we considered unnecessary to remove any items from the questionnaire to improve internal consistency Additionally, we estimated split-half reliability using Spearman-Brown’s ρ. Values obtained were 0.801 for the test dataset and 0.839 for the retest dataset, which indicate an acceptable internal consistency (>0.80).

#### 3.4.2. Stability

Since the questionnaire was given twice to four different groups of people, we started by running normality tests on the questionnaire scores for each individual group and test case independently (8 subgroups in total). We applied Shapiro–Wilk over each subgroup scores taking a significance level of α = 0.05. In all eight cases, the resulting *p*-value was higher than α, thus reassuring the normality of the data and encouraging the use of parametric methods for the subsequent analysis. We checked homoscedasticity between test and retest sets considering separate groups and using Levene’s test.

Next, we conducted a two-tailed paired sample *t*-test to ensure that there were no significant differences between the test and retest score sets ($H_{0}:\mu_{d}=0,H_{1}:\mu_{d}\neq 0$). The result supported the null hypothesis, as *p* > 0.05 (*p* = 0.243), although we do observe a slight increase of average scores in retest. Repeating the analysis by individual groups, the resulting *p*-values ranged from 0.237 (UB-NHD, major observable difference) to 0.956 (CSK, minor variation). We detected no significant differences in test and retest scores for each group of subjects, but we observed a slight consistent increase of scores in each group (Table 3).

The global sets of scores yielded to a strong Pearson correlation between individuals on the two attempts (r=0.895,p<0.001,n=174). Pearson’s *r* > 0.7 indicates adequate test–retest reliability, and thus a high repeatability of results (Figure 3). When computing correlations for each group, we obtained statistically significant Pearson correlation coefficients: FCB (*r* = 0.626, *p* < 0.001, *n* = 36), UB-NHD (r=0.529,p=0.003,n=31), CSK (r=0.706,p<0.001,n=77) and UB-FIL (r=0.805,p<0.001,n=30).

## 4. Discussion

Validated tools for assessing nutrition knowledge among athletes are needed, with the final aim of investigating the relationship between nutrition knowledge and dietary habits in this population [15]. This study reports the development and validation of a reliable and compact sports nutrition questionnaire in both English and Spanish. The questionnaire was validated for its content, construct, reliability and feasibility. To our knowledge, this is the first study to develop and validate a short sports nutrition questionnaire in both English and Spanish.

The final questionnaire includes 24 questions that comprise 59 items. It is divided into four sections: macronutrients (30 items), micronutrients (19 items), hydration (7 items) and food intake periodicity (3 items). The questionnaire can be answered in 12 min on average and is centered on issues that are considered priorities in the education of elite athletes. It does not include aspects such as supplements intake, weight control and alcohol consumption, because they are subjected to nutritionist advice, discouraged or forbidden for athletes (e.g., alcohol consumption). Questions are based on recent guidelines and have been thoroughly revised and adjusted according the commentaries received from experts. On the contrary of other similar validated questionnaires on sports nutrition knowledge [15], NUKYA is considerably shorter and requires less time to be completed. Moreover, it has been validated for a range of ages (13 to more than 25 years) that includes adolescents, paying special attention to the wording and terms used. We assessed the reliability and validity of our questionnaire using a robust methodology and a large sample of individuals with predictably different levels of sport nutrition knowledge. Moreover, to assure the quality of the responses, we always supervised the administration of the questionnaire. Nothing suggests that the questionnaire cannot be successfully administered to adults older than those participating in the study.

The NUKYA questionnaire was developed through an iterative process of consultation with different groups of experts. First, the questionnaire was validated at the design stage, and then during content validation, using the CVI and obtaining excellent content validity indices. As Trakman et al. [19] pointed out, quantitative methods such as CVI are effective but unusual when validating the content of sports nutrition knowledge questionnaires. Experts from the panel that validated the content and respondents of the pilot study agreed that the questionnaire covered the concepts it intends to measure, so that face validity seemed to be fulfilled. The feasibility also seems adequate according to the low administration time and the really few demands for clarification from participants.

The questionnaire had satisfactory construct validity, which was ascertained with different methods: (i) a comparison of groups who were expected to obtain different scores; (ii) an analysis of item difficulty and discrimination; (iii) a sectional analysis; and (iv) a Rasch analysis, a model developed according the item–response theory. In the comparison of groups, nutrition students obtained the best scores, followed by university students with no formal studies in health and, finally, by athletes and high school students. Differences between groups were statistically significant in some cases and coherent with the formal education received and age of individuals. In most cases, in the sectional analysis, we observed the same patterns. A few alterations are detected, such as hydration and periodicity of food intake, and they are easily explained. The analysis of item difficulty and discrimination detected six items that could be considered too easy or too difficult. However, these items were retained because they all showed a good discrimination index and were considered important to assess if certain concepts were present in the knowledge structure of the subjects. Additionally, Rasch analysis indicated that participants who performed well throughout were more likely to answer particular questions correct. The person–item map showed a good superposition between item difficulty range and the person measure distribution, so we can state that the test meaningfully measures the ability of all respondents.

Regarding reliability, internal consistency was examined through the Cronbach’s alpha and the split-half methods, both resulting in a good item consistency. Cronbach’s alpha was recalculated by extracting each item once, and values were equally high. Thus, such evidence indicates that items included in the questionnaire are measuring the same underlying concept. Moreover, questionnaire stability was assessed by comparing test and retest scores, and no significant differences were detected between them, either globally or group by group. Finally, the correlation between test and retest was high, and we also obtained significant correlations for each group. These data indicate that the questionnaire has an excellent repeatability. We detected a non-significant but constant increase in retest scores compared to test scores, across the four subject groups. This effect is well described elsewhere [28] and can be attenuated selecting a long test–retest interval. In our case, the interval was 15 days, whereas an interval of 2 or 13 days is generally recommended. Other factors that can influence this retest effect are test form, test modality and participants’ age.

In their systematic review, Trakman et al. [15] developed and used two ratings for sports nutrition knowledge questionnaires. One rating was used to assess comprehensiveness and included 11 items (general knowledge, carbohydrates, protein, fat, micronutrients, pre-exercise, post-exercise, during exercise, recovery, fluid, supplements and alcohol). The second rating was used to assess validity and reliability evidence (face validity, content validity, item discrimination, internal reliability, construct validity and external validity). These ratings were based on the guidelines developed by Parmenter and Wardle [17]. In both cases, one point was awarded for each item covered, with a maximum score of 11 and 6, respectively. Applying these ratings to our questionnaire, the NUKYA questionnaire obtained nine points in comprehensiveness and six in validity and reliability scores, which are in the upper range of those published in the systematic review. We were also able to give a positive response to the complementary question of whether a pilot study was conducted.

The NUKYA questionnaire has also several limitations. The most important one is that it does not include questions on supplements intake, weight management and alcohol consumption. On the one hand, this can be a drawback in certain cases. On the other hand, we obtain a quick assessment of the nutritional knowledge of aspects related to the maintenance of a balanced diet and its adaptation to sport practice. Another limitation is that we did not assess the validity of the questionnaire among different nationalities, although our sample included a small percentage of subjects from foreign countries. Additionally, some items of the questionnaire have high (*n* = 4) or extremely low (*n* = 2) difficulty indices. Likewise, Wald test showed that some items (*n* = 5) could be removed without influencing the model. However, as the test performance is not compromised, these items were retained because they assessed concepts that were considered important in the nutrition education of athletes. The NUKYA questionnaire has been validated for team sport athletes and for Spanish population, although some of the subject groups included individuals from other nationalities. Consequently, the questionnaire would have to be re-validated before being used in individual sport athletes or in other populations. Finally, we were not able to differentiate and compare the responses obtained from athletes in the English and Spanish versions of the questionnaire. This was because the online form allowed athletes to choose the language they felt more comfortable with, and the answers were recorded in the same database without any differentiation.

## 5. Conclusions

We developed a questionnaire in English and Spanish to assess sports nutrition knowledge among team sport athletes older than 12: NUKYA. We validated the questionnaire according to the classical test theory (CTT) and the item responses theory (IRT). For the latter, we used the Rasch model with the extension rating scale model, on a large sample of individuals. Statistical analyses of content validity and construct validity (test–retest reliability and internal consistency) prove that NUKYA is a valid and reliable medium-length questionnaire. It can be used to evaluate nutrition knowledge in both amateur and professional team sport athletes. It may help to identify knowledge gaps and plan educational interventions (e.g., for sports clubs and university sports teams). Finally, it is possible to study the influence of nutritional education on dietary habits using this test and assess/monitor the sports nutrition knowledge in large populations through electronic forms.

## Figures and Tables

**Figure 1 nutrients-12-03561-f001:**
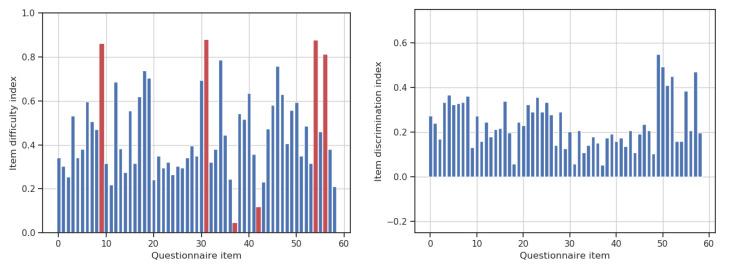
(**Left**) Item difficulty indices for each questionnaire item. Ninety percent of the items are within the range (0.2,0.8). Items outside of that range are highlighted in red. (**Right**) Item discrimination indices for the same items. All questions positively discriminate high-scored respondents from low-scored ones.

**Figure 2 nutrients-12-03561-f002:**
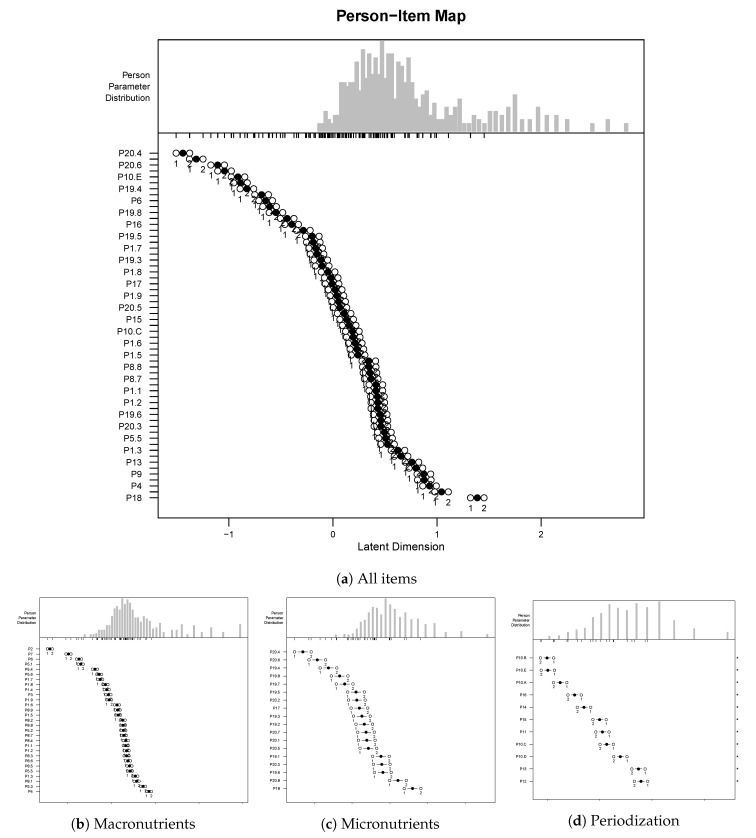
Person–item maps for the fitted rating scales model. (Top) The questionnaire items (y-axis) sorted according to the value they take in the latent dimension (x-axis, negative to positive means easier to more difficult). (Bottom) The person measure distribution. Items should ideally be located along the whole scale to meaningfully measure the ‘ability’ of all persons.

**Figure 3 nutrients-12-03561-f003:**
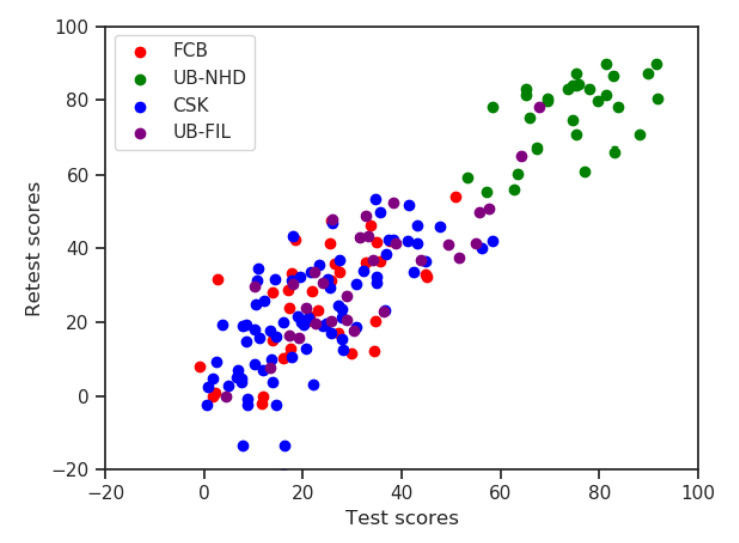
Scatter plot of questionnaire scores by group.

**Table 1 nutrients-12-03561-t001:** Characteristics of the groups participating in the questionnaire validation study. Numbers for each group expresses frequencies and percentages.

		FCB	UB-NHD	CSK	UB-FIL
Test set	n=218	37	49	93	39
Retest set	n=173	36	30	77	30
Rasch set	n=445	264	49	93	39
Gender (Rasch set)	Male	234 (88.6%)	15 (30.6%)	42 (45.2%)	27 (69.2%)
	Female	30 (11.4%)	34 (69.4%)	51 (54.8%)	12 (30.8%)
Age (Rasch set)	13–15	80 (30.3%)	–	88 (94.6%)	–
	16–18	88 (33.3%)	–	5 (5.4%)	1 (2.6%)
	19–21	32 (12.1%)	2 (4.1%)	–	7 (17.9%)
	22–25	23 (8.7%)	34 (69.4%)	–	16 (41.0%)
	>25	18 (6.8%)	13 (26.5%)	–	15 (38.5%)

FCB, elite athletes; UB-NHD, final year students of nutrition; CSK, high-school students; UB-FIL, second year students of philosophy.

**Table 2 nutrients-12-03561-t002:** I-CVI statistics and S-CVI values for the initial and reviewed set of questionnaire items. Changing values are in bold.

			Relevance	Clarity	Ambiguity	Simplicity
	I-CVI	min	0.86	0.71	0.86	0.71
Initial item set		max	1.00	1.00	1.00	1.00
(n=26)		$\bar{x}\pm s$	0.99±0.04	0.97±0.07	0.99±0.04	0.97±0.07
	S-CVI	–	0.92	0.81	0.92	0.81
	I-CVI	min	0.86	**0.86**	0.86	**0.86**
Reviewed item set		max	1.00	1.00	1.00	1.00
(n=24)		$\bar{x}\pm s$	0.99 ± 0.04	**0.98** ± **0.05**	0.99 ± **0.03**	**0.98** ± **0.05**
	S-CVI	–	0.92	**0.88**	**0.96**	**0.88**

**Table 3 nutrients-12-03561-t003:** Test and retest statistics of the four groups of subjects included in this validation study. It is apparent that UB-NHD ranks consistently higher on nutrition knowledge.

Group				95% CI	
		Average $\bar{x}$	Std Dev *s*	Lower Bound	Upper Bound
UB-NHD	Test	74.4	10.0	54.4	94.4
UB-NHD	Retest	76.6	10.1	56.5	96.7
UB-FIL	Test	33.4	15.9	1.6	65.2
UB-FIL	Retest	34.4	16.5	1.3	67.5
FCB	Test	23.3	12.2	−1.1	47.7
FCB	Retest	25.3	14.6	−4.0	54.6
CSK	Test	21.7	13.4	−5.0	48.5
CSK	Retest	21.8	16.3	−10.7	54.4

**Table 4 nutrients-12-03561-t004:** Averages and standard deviations ($\bar{x}\pm s$) for sectional normalized scores, for each group and globally.

	Macronutrients	Micronutrients	Hydration	Food Intake Periodicity
CSK	15.6±16.3	32.1±22.3	26.2±16.5	32.1±33.2
FCB	20.9±18.8	28.7±19.8	38.7±23.2	58.6±36.1
UB-FIL	27.9±23.7	41.4±20.5	28.6±19.1	45.7±29.7
UB-NHD	78.3±15.5	71.4±16.1	62.6±29.9	93.2±18.1
Global	29.2±27.3	36.7±24.1	39.3±25.0	57.4±37.6

**Table 5 nutrients-12-03561-t005:** Fitting results for the rating scales model. Higher separation reliability values represent better model fitting and a reliability below 0.50 indicates that the differences between measures are mainly due to measurement error. A single global model explains the data better than multiple sectional models.

	Separation	Observed	Mean Square
	Reliability	Variance	Measurement Error
All items	0.8610	0.2677	0.0372
Macronutrients	0.8577	0.5368	0.0818
Micronutrients	0.6416	0.3659	0.1297
Inner-periodization	0.6398	0.5779	0.2122

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
