# Peer review of "Development and Validation of a Short Sport Nutrition Knowledge Questionnaire for Athletes"

_nutrients, 2020, doi:10.3390/nu12113561_

Round 1

Reviewer 1 Report

Consider looking at the following publication that came out in 2019 in the US as another reference and justification for the need for a new instrument. 

Sports Nutrition Knowledge of Certified Athletic Trainers by 
Shelley L. Holden, Neil A. Schwarz, and Geoffrey M. Hudson

Line 85 typo "cvareer" instead of career.

Author Response

We would like to thank the reviewer for their comments and suggestions for our manuscript. Below, we provide a  description  of  how  each  those comments  were  addressed in  our  manuscript. 

Consider looking at the following publication that came out in 2019 in the US as another reference and justification for the need for a new instrument.

Sports Nutrition Knowledge of Certified Athletic Trainers by Shelley L. Holden, Neil A. Schwarz, and Geoffrey M. Hudson

Answer: We have incorporated this reference to the introduction section.

 Line 85 typo "cvareer" instead of career

Answer: We have corrected this mistake.

Reviewer 2 Report

The authors performed an interesting and well-prepared work which result (the NUKYA questionnaire) may be considered as a useful tool in the assessment of nutritional knowledge of athletes. However, the work contains some aspects listed below that should be corrected before further proceeding.

  1. The full version of the NUKYA questionnaire has not been made available for review,
    which prevents the final evaluation of the work.
    2. Consider whether at work the term "nutritional knowledge" should not be used instead of
    "nutrition knowledge".
    3. It would also be worth pointing out in the work (e.g. discussion or conclusions) that this
    questionnaire is particularly specific (due to the groups studied) for team sports.
    4. Line 28 – “expertsclaim” - these words should be separated by a space.
    5. Line 65 - logically the past tense should be used here ("are" -> "were").
    6. Line 75 - correct "scgreening" to "screening".
    7. Line 79 - logically the past tense should be used here ("is" -> "was").
    8. Line 85 - correct "cvareer" to "career".
    9. Line 93 – why "and" was written in italics.
    10. Line 160 - logically the past tense should be used here ("are" -> "were").
    11. Line 169 - remove the space before the dot at the end of the sentence.
    12. Line 193 - the inscription "p" without the addition "- value" is justified in parentheses.
    13. Line 216 - remove an extra dot.
    14. In figure 2 the description "(a) All items" overlaps another notation.  15. Line 293 - consider using a slightly more formal word than "huge".
    16. Line 318 - it seems that the sentence started with "Person-item map" is "truncated" and not finished.
    17. Line 368 – consider rewording "we can study the influence” in the impersonal direction, like "it is possible ..”.
    18. References - Editorial writing should be standardized. Journal names should be spelled correctly in the context of upper / lower case letters.

Author Response

The authors performed an interesting and well-prepared work which result (the NUKYA questionnaire) may be considered as a useful tool in the assessment of nutritional knowledge of athletes. However, the work contains some aspects listed below that should be corrected before further proceeding.

Answer: We would like to thank the reviewer for their comments and suggestions for  our manuscript.  We are convinced that our manuscript is now a more consistent and clearer piece of research thanks to their feedback.  Below, we provide a point by point  description  of  how  each  those comments  were  addressed in  our  manuscript.

1.The full version of the NUKYA questionnaire has not been made available for review, which prevents the final evaluation of the work.

Answer: We have attached the questionnaire in this first revision round, and we will made it available as a supplementary material.

  1. Consider whether at work the term "nutritional knowledge" should not be used instead of "nutrition knowledge".

Answer: We have changed “nutritional knowledge” to “nutrition knowledge”.  

  1. It would also be worth pointing out in the work (e.g. discussion or conclusions) that this questionnaire is particularly specific (due to the groups studied) for team sports.

Answer: We have incorporated two sentences in both discussion and conclusions sections (lines 354 and 367) that point out this issue.

  1. Line 28 – “expertsclaim” - these words should be separated by a space.
  2. Line 65 - logically the past tense should be used here ("are" -> "were").
  3. Line 75 - correct "scgreening" to "screening".
  4. Line 79 - logically the past tense should be used here ("is" -> "was").
  5. Line 85 - correct "cvareer" to "career".
  6. Line 93 – why "and" was written in italics.
  7. Line 160 - logically the past tense should be used here ("are" -> "were").
  8. Line 169 - remove the space before the dot at the end of the sentence.
  9. Line 193 - the inscription "p" without the addition "- value" is justified in parentheses.
  10. Line 216 - remove an extra dot.

Answer: All mistakes in wording have been corrected.

  1. In figure 2 the description "(a) All items" overlaps another notation.

Answer: Originally, manuscript have been submitted in LaTeX format. When it was converted to PDF the figure looked good. However, when the PDF was converted to the Word format, some figures were out of the frame. We have tried to correct this, but it was impossible. However, in the PDF file all figures are correctly included.

  1. Line 293 - consider using a slightly more formal word than "huge".

Answer: We have changed “huge” to “large”

  1. Line 318 - it seems that the sentence started with "Person-item map" is "truncated" and not finished.

Answer: This sentence has been completed.

  1. Line 368 – consider rewording "we can study the influence” in the impersonal direction, like "it is possible ..”.

Answer: We have changed the sentence to impersonal direction.

  1. References - Editorial writing should be standardized. Journal names should be spelled correctly in the context of upper / lower case letters.

Answer: We have checked the references included in the manuscript and have corrected them.